# Enzymatic Bioreactors: An Electrochemical Perspective

**Simin Arshi [1]**, **Mehran Nozari-Asbemarz [2]** and **Edmond Magner [1,*]**

[1] Department of Chemical Sciences and Bernal Institute, University of Limerick, V94 T9PX Limerick, Ireland; simin.arshi@ul.ie

[2] Department of Chemistry, University of Mohaghegh Ardabili, Ardabil 5619911367, Iran; m.nozari@uma.ac.ir

[*] Correspondence: edmond.magner@ul.ie; Tel.: +353-61-202629

**Abstract:** Biocatalysts provide a number of advantages such as high selectivity, the ability to operate under mild reaction conditions and availability from renewable resources that are of interest in the development of bioreactors for applications in the pharmaceutical and other sectors. The use of oxidoreductases in biocatalytic reactors is primarily focused on the use of NAD(P)-dependent enzymes, with the recycling of the cofactor occurring via an additional enzymatic system. The use of electrochemically based systems has been limited. This review focuses on the development of electrochemically based biocatalytic reactors. The mechanisms of mediated and direct electron transfer together with methods of immobilising enzymes are briefly reviewed. The use of electrochemically based batch and flow reactors is reviewed in detail with a focus on recent developments in the use of high surface area electrodes, enzyme engineering and enzyme cascades. A future perspective on electrochemically based bioreactors is presented.

**Keywords:** bioelectrocatalysts; oxidoreductases; biocatalytic reactors; electrochemical reactors

## 1. Introduction

Biocatalysts represent an alternative to conventional catalysts, providing a number of advantages that include availability from renewable resources, biodegradability and high selectivity [1]. Enzymes are proteins and catalyse a wide variety of reactions that have applications in a range of industrial processes [2,3]. They are extremely effective biological catalysts, highly selective and can operate under mild conditions (ambient temperatures, physiological pH and atmospheric pressure). When used in organic synthesis, enzymes can obviate the need to protect and activate functional groups. Enzymes are generally soluble in water and can avoid the use of organic solvents, resulting in the generation of less waste [4–6]. Oxidoreductases are a class of enzymes that catalyse redox reactions [7] such as oxygenation, dehydrogenation, oxidative bond formation and electron transfer reactions [8]. Over the last four decades, oxidoreductases have been coupled successfully with electrochemistry in biofuel cells [9–12] and biosensors [13–16]. The use of oxidoreductases in biocatalysis has gained significant attention in the synthesis of a variety of chemicals, such as chiral compounds [17–19], biofuels [20] and ammonia [21]. While biocatalysts are most commonly used in batch reactors, the development of biocatalytic flow reactors is of increasing interest as it can bring significant advantages that include improved mass and heat transfer, lower costs and higher yields [22,23]. This review describes oxidoreductases and bioreactors from an electrochemical perspective, with a focus on the electrocatalytic activity of oxidoreductases and methods to improve their use.

## 2. Bioelectrochemistry and Bioelectrocatalysts

Oxidoreductases undergo electron transfer with the surface of an electrode via two mechanisms [24]; mediated [25–28] and direct electron transfer [29–31] (Figure 1). As the redox active centres of

oxidoreductases can be placed deep within the enzymes, the rate of electron transfer between the redox centre and the electrode can be restricted and only a limited number of redox enzymes can undergo direct electron transfer (DET) [32]. Direct electron transfer relies on the appropriate orientation of an enzyme on the electrode surface, the location of the active site within the enzyme and the distance between the redox centre and the electrode. Efficient rates of electron transfer require that the redox active site in the enzyme be close to the surface of the electrode, with the appropriate orientation of the enzyme on the electrode. When the redox site is inaccessible, mediators can be used as electron shuttles between the electrode and the active site of enzymes (Figure 1a) [33,34].

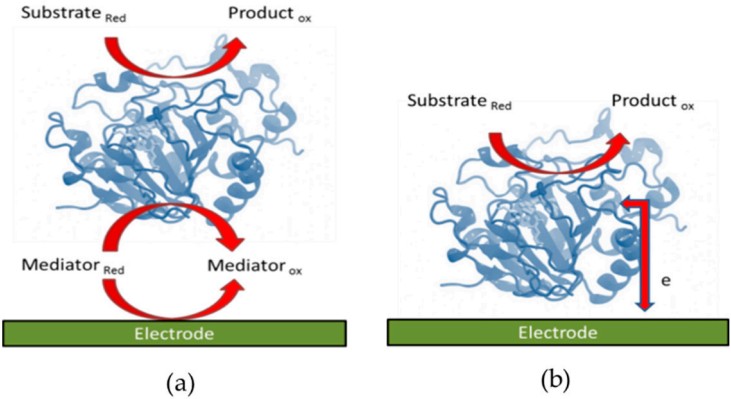

**Figure 1.** Schematic diagram of (**a**) mediated and (**b**) direct electron transfer between an enzyme and an electrode.

*2.1. Mediated Electron Transfer*

The redox properties of the mediator and of the enzymes need to match in order to have an efficient rate of electron transfer. The reduction potential of the mediator ($E^\circ_{mediator}$) should be more positive than the redox potential of the prosthetic group of the enzyme ($E^\circ_{enzyme}$) for oxidation to occur, while for reduction, $E^\circ_{mediator}$ should be more negative than $E^\circ_{enzyme}$. The mediator should undergo fast electron transfer both at the surface of the electrode and with the active site of the enzyme [34,35]. Properties such as mediator stability, selectivity and the electrochemical reversibility of the redox couple also need to be considered [33]. Approximately 90% of oxidoreductases utilised nicotinamide coenzymes to catalyse reactions. For example, alcohol dehydrogenase (ADH) uses the cofactor nicotinamide adenine dinucleotide ($NAD^+$) to catalyse the oxidation of substrates such as cyclohexanol [36], primary and secondary alcohols [37] and methionine [38]. Due to the high cost of $NAD(P)^+$, it is important to use an efficient regeneration system. The direct regeneration of nicotinamide coenzymes at unmodified electrode surfaces requires the use of high overpotentials that can result in the formation of dimers of $NAD(P)^+$ that are enzymatically inactive. Mediators such as methylene green, methylene blue [39–42], 2,2-azino-bis-(3-ethyl-benzo-thiazoline-6-sulfonic acid (ABTS) [43], naphthoquinone [44], ferrocenes [45,46], and viologens [47] are widely used to regenerate $NAD^+$, etc. For example; an indirect $NAD(P)^+$ regeneration system utilised ABTS to oxidise alcohols with a turnover of 1200 h$^{-1}$ [48]. A number of enzymes such as dihydrolipoamide dehydrogenase (DLD), ferredoxin-NADP$^+$ reductase (FNR) have been used to regenerate NAD [49–51]. Chen et al. used reduced methyl viologen ($MV^{\bullet+}$) and diaphorase for effective NADH regeneration in the production of $NH_3$ and for asymmetric amination [52]. The mediated electrochemical regeneration of NADH through methyl viologen was also used in the enzymatic reduction of ketones [53]. Badalyan et al. used the negatively charged viologen derivative [(SPr)$_2$V$^{\bullet}$]$^-$ as an effective mediator for nitrogenase electrocatalysis [54]. Tosstorff et al. used three different mediators, cobalt sepulchrate, safranin T, and [Cp*Rh(bpy)(H$_2$O)]$^{2+}$ in an NADH regeneration system that was combined with old yellow enzyme for the asymmetric reduction of C=C [55].

Redox polymers are used to wire enzymes and also act as an immobilisation matrix facilitating electron transfer between the electrode and enzyme by transferring electrons within the polymer matrix. Redox polymers possess a polymeric backbone with attached redox mediators [56,57]. Early reports focused on redox polymers based on osmium complexes with polymer backbones comprised of poly(vinyl imidazole)s and poly(vinyl pyridine)s [58,59]. A range of other redox polymers based on poly(vinylalcohol), poly(vinyl imidazole), poly(vinyl pyridine) and poly(ethylenimine) modified with redox species such as ferrocene [60], cobaltocene [61], viologen [62] and quinone [44] have been described. For example, Alkotaini et al. used a redox polymer N-benzyl-N'-propyl-4,4'-bipyridinium-modified linear polyethylenimine benzylpropylviologen (BPV-LPEI) and diaphorase for effective cofactor regeneration for the bioelectrosynthesis of the bioplastic, polyhydroxybutyrate (PHB) [62]. Szczesny et al. used a viologen-modified redox polymer for the electrical wiring of W-dependent formate dehydrogenase to reduce $CO_2$ gas to formate [63], whereas Yuan et al. described the electrochemical regeneration of NADH using a cobaltocene-modified poly(allylamine) redox polymer and diaphorase. The system produced 1,4-NADH with high yields close to 100% and turnover frequencies between 2091 and 3680 $h^{-1}$ at different temperatures. The system was coupled with ADH to produce methanol and propanol (Figure 2) [57].

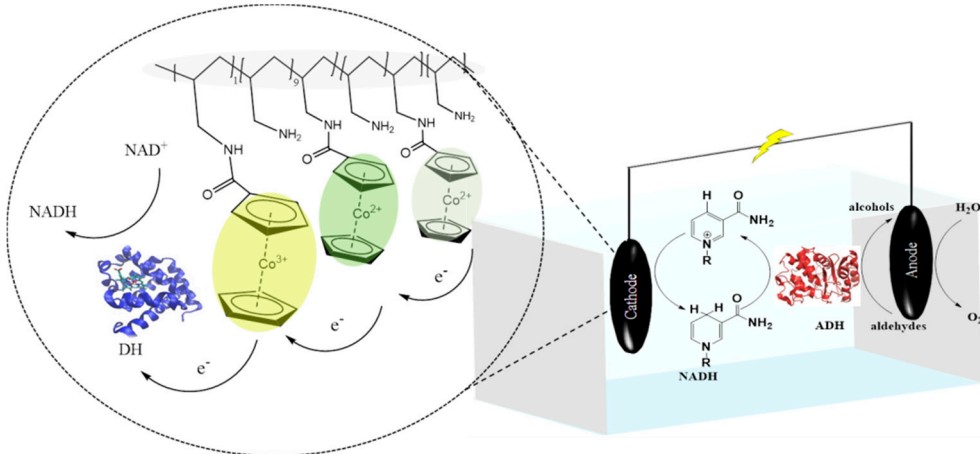

**Figure 2.** A schematic of the designed system for the electrochemical regeneration of NADH [57].

*2.2. Direct Electron Transfer*

Direct electron transfer has a significant advantage in comparison to mediated electron transfer [64] as the use of mediators can cause voltage losses stemming from the difference between the redox potentials of the enzyme and the mediator. In addition, the development and miniaturization of biofuel cells can be easier as no membrane or compartment is required [51,64,65]. Some oxidoreductases, such as multi copper oxidases, can undergo direct electron transfer with the electrode. In direct electron transfer, the distance (d) between the electrode and cofactor plays a key role in the rate of electron transfer. In an electron transfer reaction, the heterogeneous rate constant k° depends on the distance between the mediator and the electrode, (Equation (1)) [66,67]:

$$k° = k°_{maximum} \exp(-\beta d) \tag{1}$$

where β is the decay coefficient, d is the distance between the redox centre of the enzyme and the electrode and $k°_{maximum}$ is the rate constant at the closest approach. The orientation of enzymes on the electrode surface can affect d so it is crucial that enzymes have the optimal orientation on the surface for direct electron transfer [67]. In general, the distance should be less than 2 nm. At longer distances, the rate of electron transfer between the electrode and cofactor is too low [35]. DET studies of multi-copper oxidases such as laccase, cellobiose dehydrogenase and pyrroloquinoline quinone-dependent glucose dehydrogenases have been widely reported [34]. Interprotein electron

transfer occurs readily in multi-copper oxidases, minimizing the distance for electron transfer and increasing the rate of electron transfer [68]. For example; the copper $T_1$ site is the primary electron acceptor site and then transfers electrons via an intermolecular mechanism to other copper sites, where $O_2$ is reduced [69]. The greatest challenge in direct electron transfer is to enable electrical communication between the electrode and enzyme. Some methods of enzyme immobilisation can improve DET [35]. For example, Lee et al. used a gold-binding peptide (GBP) to bind glucose dehydrogenase which facilitates enzyme orientation on the surface of the gold electrode, decreasing the distance between flavin adenine dinucleotide (FAD) and the electrode surface [70]. Cross-linked hydrogels can be used for enzyme immobilisation on the surface of an electrode. Liu et al. used an agarose hydrogel to immobilise haemoglobin, myoglobin and horseradish peroxidase on the electrode surface whereby the proteins could undergo direct electron transfer [71]. Kuk et al. reported the electroenzymatic reduction of $CO_2$ using NADH-free formate dehydrogenase immobilised on a conductive polyaniline hydrogel. The hydrogel amplified the electron transfer between the electrode and enzyme and moreover, it increased enzyme loading [72]. Hickey et al. reported successful direct electron transfer between enzymes (laccase and nitrogenase) and an electrode using carbon electrodes modified with multi-walled carbon nanotubes and poly(ethylenimine) attached to pyrene moieties. The system was used for the catalytic reduction of $O_2$ and the production of $NH_3$ from $N_2$ [73]. Much of the work on the direct electron transfer of redox enzymes has focused on biosensors and biofuel cells as DET offers a number of advantages. DET enables the use of reagentless biosensors, an important advantage for such devices [74] while with biofuel cells, the absence of a mediator reduces voltage losses arising from differences in the redox potentials of the mediator and the redox enzyme.

## 3. Immobilisation Strategies

There are five general methods of immobilising enzymes; physical adsorption, covalent binding, immobilisation via ionic interactions, cross linking and entrapment in a polymeric gel or capsule [75]. Physical adsorption is the easiest method of immobilisation [33,34]. Sakai et al. used a glassy carbon electrode modified with 4-mercaptopyridine and gold nanoparticles to adsorb formate dehydrogenase for the oxidation of $HCOO^-$ oxidation and the reduction of $CO_2$ [76]. However, due to weak interactions, physisorbed enzymes can leach from the electrode surface. Covalent binding onto electrode surfaces provides very stable enzyme immobilisation [75,77]. The most commonly used electrodes in this method are gold and carbon electrodes [34]. The immobilisation of bilirubin oxidases on nanoporous gold electrodes was performed by attaching -COOH via diazonium surface coupling to the electrode surface which was then used for the covalent coupling of bilirubin oxidases [78]. Immobilisation via ionic interactions is commonly used and is dependent on parameters such as pH and the concentration of salt [75]. The crosslinking of enzymes occurs via a bifunctional agent such as glutaraldehyde which results in enzyme aggregation, with the enzymes acting as their own carrier [75]. Encapsulation techniques entail the entrapment of enzymes in the pores of polymers, hydrogels and sol–gels. Polymers are crossed linked in the presence of enzymes, encapsulating the enzymes in the polymers [34,79]. Redox polymers (Section 2.1) were widely used for the immobilisation of enzymes, with immobilisation occurring via electrostatic interactions, cross linking or/and encapsulation. These polymers are used for the construction of cathodes and anodes in biofuel cells as they enable the successful electrical connection of enzymes, providing a stable means of attachment and can be miniaturized [80,81]. On account of these advantages, redox polymers are successfully and widely used for the construction of biofuel cells [82], biosensors [83], biosupercapacitors [84] and bioelectrosynthesis [85]. As an example of a biocatalytic system, Alsaoub et al. constructed a biosupercapacitor using an Os complex modified polymer to immobilize glucose oxidase and flavin adenine dinucleotide (FAD)-dependent glucose dehydrogenase at the anode and bilirubin oxidase at the cathode [86].

## 4. Biocatalytic Reactors

Biocatalytic reactors consist of one of two types of reactor, batch and flow reactors [87,88].

### 4.1. Batch Reactors

Batch reactors are simple and flexible in terms of manufacture. They can vary in size from small (mL) to large (m$^3$). Immobilised enzymes can be recovered and reused in batch reactors, while additional amounts of enzyme can be added if required [87,89]. Markle et al. developed a batch reactor in which deracemization, stereoinversion and the asymmetric synthesis of l-leucine were carried out by combining enzymatic oxidation using d-amino acid oxidase with an electrochemical reduction step [90]. Mazurenko et al. used membrane-bound (S)-mandelate dehydrogenase encapsulated in silica film on an electrode surface to prepare phenylglyoxylic acid from an (S)-mandelic acid in a batch reactor [91]. Ali et al. developed an electrochemical batch reactor for the regeneration of 1,4-NADH via platinum and nickel nano-particles deposited on the electrode surface [92]. While batch systems are in widespread use in industrial applications, the development of flow reactors has been the subject of significant research as outlined below [93].

### 4.2. Flow Reactors

The use of flow can increase rates of mass transfer with subsequent increases in the rates of reaction, decreasing the reaction time and making it feasible to perform reactions at a large-scale using relatively small scale equipment [89]. Kundu et al. constructed a packed microreactor to investigate the polymerization of polycaprolactone from $\varepsilon$-caprolactone in batch and continuous flow modes. Polymerization in continuous flow mode was faster than in batch mode and higher molecular mass polymers were obtained in continuous flow mode [94]. Using lipase, the time for the resolution of ($\pm$)-1,2-propanediol decreased significantly from 6 h (batch) to 7 min (flow) [95]. Using enzymes as a catalyst for the production of chemicals on a large scale is limited due to the requirement for high concentrations of enzyme, a limitation that does not necessarily apply to flow reactors [96]. For example, Cosgrove et al. used galactose oxidase to investigate the oxidation of lactose on a large scale in batch and flow systems. In the batch reactor rate of reaction of 0.74 mmol L$^{-1}$ h$^{-1}$ was reported, a rate that increased 224-fold in the flow reactor (167 mmol L$^{-1}$ h$^{-1}$). Moreover, the concentration of enzyme decreased considerably from 2.5 mg mL$^{-1}$ (batch reactor) to 0.5 mg mL$^{-1}$ (flow reactor) [96]. Flow reactors possess additional advantages over batch reactors such as better mixing and thermal control, improved stability and life time [23,97]. A range of studies have described the use of flow reactors to prepare intermediates for the preparation of pharmaceutical drugs [98], agrochemicals such as fluorinated and chlorinated organic compounds [99] and electronic materials such as iron silicide–carbon nanotubes used in Li-ion batteries [100,101]. A flow reactor is based on the principle of using a pump to pump the substrate into a reactor where a product is formed and is then pumped out of the reactor. Pumps can be divided into two groups, continuous and semi-continuous systems. Semi-continuous systems need to be refilled while continuous flow systems do not require refilling. Generally, reactors are fabricated from glass, polymers (e.g., polytetrafluoroethylene, polyfluoroacetate and polyether ether ketone), stainless steel and metals [23,102].

In general, enzyme immobilisation brings about a number of benefits. Lower amounts of enzyme are required, making the process more cost effective [103], while the removal of product from the reactor is simple and easy. In addition, immobilised enzymes show high stability, selectivity in comparison with free enzymes [23]. As described earlier, there are five general methods of immobilising enzymes. A number of challenges can arise when enzymes are immobilised in a microreactor. The enzymes need to be stable under conditions of fluid flow. In addition, during the immobilisation process, clogging and solid formation should be minimized [104]. Enzymes can be immobilised in flow reactors in a number of ways. Enzymes can be attached to beads, loaded into the reactor. This approach can result in high enzyme loadings but requires high back pressures [105]. The enzyme can be immobilised on the walls of reactors [89,106,107], attached to inorganic and polymeric monoliths in the channels [108–110] immobilised on a membrane (a method that is outside the scope of this paper) [111–114].

In order to select the optimal conditions for the immobilisation of an enzyme, the support needs to be selected. The type of support chosen will determine the immobilisation procedure [106]. The process

of enzyme immobilisation should be cost-effective and technologically efficient. Flow reactors can be divided into three groups, packed-bed microreactors, monolithic microreactors and wall-coated microreactors (Figure 3).

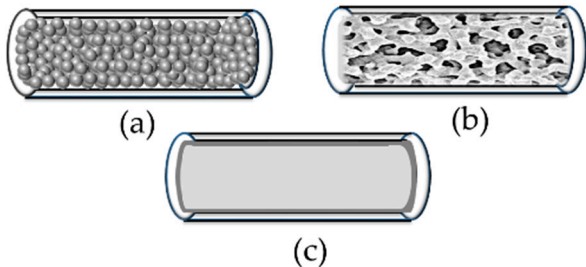

**Figure 3.** A schematic of (**a**) packed-bed; (**b**) monolithic; and (**c**) wall-coated microreactors.

### 4.2.1. Packed-Bed Reactors

Conventional packed-bed reactors suffer from disadvantages such as inefficient heat and mass transfer due to the use of large particles (mm in size) in the channels [97]. The diameter of the particles should be less than 1/20th of the channel diameter in order to prevent channelling and ensure even flow solution. For microreactors, the diameters of the particles should be less than 50 nm [115]. Enzymes can be immobilised on porous beads, streptavidin-coated magnetic microbeads, porous resins or various hydrogels. Oglio et al. designed a pack-bed flow reactor using immobilised ketoreductase and glucose dehydrogenase on aldehyde agarose for different ketones reduction. Both enzymes showed good stability in DMSO and the system was used continuously for a number of weeks [116]. A packed bed bioreactor with high selectivity for the continuous production of a range of chiral alcohols was prepared by immobilising alcohol dehydrogenase fused with HaloTag$^{TM}$ on a resin containing sepharose beads [117]. A cascade reaction for the synthesis of (1S,2S)-1-phenylpropane-1,2-diol using a packed bed reactor containing immobilised fusion alcohol dehydrogenase and benzoylformate decarboxylase was described [118]. Peschke et al. immobilised (R)-stereoselective ketoreductase LbADH (alcohol dehydrogenase from Lactobacillus brevis) and glucose dehydrogenase (GDH) on magnetic beads for the preparation of (R)-alcohols. Glucose dehydrogenase was used to recycle the cofactor. The packed-bed reactor operated continuously for four days [119]. However, using beads in the reactors has disadvantages such as the need for high backpressures to achieve adequate flow rates. Moreover, it is difficult to control a multiphase flow inside the microreactor [120].

### 4.2.2. Monolithic Reactors

To address the disadvantages of packed-bed microreactors, monolithic microreactors have been prepared [93,121]. Monolithic reactors include a network of micro- or mesoporous materials that can be divided into two groups: inorganic monoliths (such as silica-based monoliths [122]) and polymeric monoliths (such as acrylamide copolymers [123] and glycidyl methacrylate copolymers [124]). The porosity of monolithic microreactors enables rapid rates of mass transfer with high flow rates possible at low pressures, resulting in improved performance when compared to conventional packed beds [93]. Biggelaar et al. immobilised ω-transaminases on silica monoliths for the enantioselective transamination reaction [125]. Szymańska et al. developed a silica monolithic flow reactor for the esterification of a primary diol by immobilising acyltransferase on silica monoliths [126]. Sandig et al. used novel hybrid monolith materials (monolithic polyurethane matrix containing cellulose beads) for the immobilisation of lipase. The use of the continuous flow resulted in an efficient reactor with a high turnover number of $5.1 \times 10^6$ [127]. Logan et al. immobilized three enzymes (invertase, glucose oxidase, and horseradish peroxidase) on polymer monoliths in different regions of a microfluidic device [128]. Qiao et al. prepared a monolithic and a wall-coated reactor, using L-asparaginase immobilised on poly(GMA-*co*-EDMA). A lower value of $K_M$ was obtained with the monolith reactor

(3.8 μM) in comparison to the wall-coated microreactor (7.7 μM) with a lower value for $V_{max}$ also reported (106.2 vs. 157 μMmin$^{-1}$, respectively), indicating that at a lower substrate concentration, a higher reaction rate was achieved in a monolithic reactor. The monolithic reactor showed a better affinity between substrate and enzyme than wall-coated reactors [129].

### 4.2.3. Wall-Coated Reactors

In wall-coated microreactors, the mass transfer resistance was decreased and fluid flow through the channel can occur without difficulties such as pressure drop and channel blocking. Fluid dynamics as well as heat transfer can be controlled more easily than in other microreactors [130]. A disadvantage of wall-coated microreactors is that lower enzyme loadings occur in comparison with other microreactors. In order to increase enzyme loadings, different strategies can be used, including the deposition of nanostructured materials such as gold nanoparticles [107], nanosprings [131], graphene oxide [132] and dopamine [133,134] on the wall as well as using multiple layers immobilised with enzymes attached to the surface of the wall [106,135,136]. For example, Valikhani et al. constructed a wall-coated microreactor with the use of silica nanosprings, comprised of helical silicon dioxide (SiO$_2$) structures grown via a chemical deposition process. The nanosprings were attached on to the channel wall and used to immobilise sucrose phosphorylase. In comparison with the unmodified surface, the loading of enzyme was significantly increased [137]. Bi et al. developed a wall-coated micro reactor in which polyethyleneimine (PEI) and Candida Antarctica lipase B were alternatively absorbed. The loading of the enzyme increased with the increasing number of layers, showing good stability and performance for the synthesis of wax ester [138]. An interesting study was carried out by Britton et al. in which enzymes were attached on the wall in specific separated zones. This type of reactor can be used for multi-step reactions for the synthesis of products such as alpha-d-glucose1-phosphate [139]. For example, Valikhani et al. described the use of sucrose phosphorylase to attach enzymes on the walls of glass microchannels for the synthesis of alpha-d-glucose1-phosphate [140].

### 4.3. Electrochemical Reactors

The coenzyme nicotinamide adenine dinucleotide is required as a co-substrate for over 300 dehydrogenases [141]. A number of studies have been carried out to use electrochemical methods to regenerate the cofactor. Yoon et al. prepared an electrochemical laminar flow microreactor to regenerate NADH in the synthesis of chiral L-lactate from the achiral substrate pyruvate (Figure 4). For this purpose, gold electrodes were deposited on the inside of a Y-shaped reactor, with two separate streams, one with buffer and the second with reagents (FAD, NAD$^+$, enzymes and substrate), with the flow directed to the cathode. The reduced cofactor, FADH$_2$ was produced at the cathode and used for the reduction of NAD$^+$ to NADH, resulting in a 41% yield of L-lactate (theoretical yield of 50%) [142].

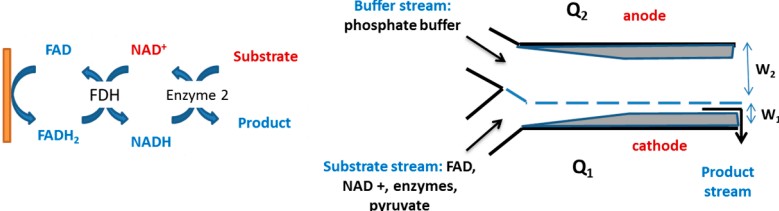

**Figure 4.** Schematic diagram of a laminar Y-shaped microreactor used for the electrochemical regeneration of NAD$^+$ using formate dehydrogenase (FDH) [142].

A filter-press microreactor with semi-cylindrical channels on the electrode surface resulting in high surface areas was used to electrochemically regenerate FADH$_2$ (Figure 5) [143].

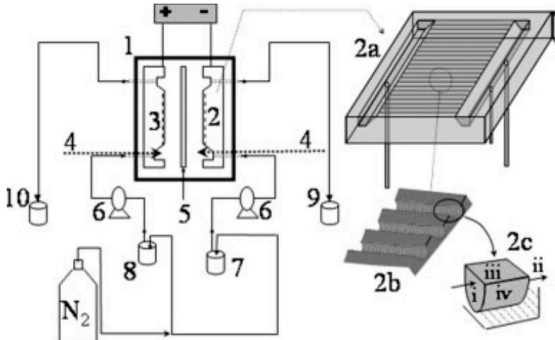

**Figure 5.** A schematic of a filter-press microreactor. Reprinted from [143] Copyright (2020), with permission from John Wiley and Sons.

As electrochemical cofactor regeneration has, by definition, to occur at the surface of the electrode, scaling up reactions is likely to cause diffusion limitations resulting in reduced rates of reaction [23]. This challenge can be addressed by using three-dimensional electrodes in continuous flow reactors. Kochius et al. designed a system for the efficient electrochemical regeneration of NAD$^+$, based on 3D electrodes with a high working surface area 24 m$^2$. The anode, comprised of a packed bed of glassy carbon particles, was surrounded by two cathodes (titanium net) (Figure 6). The oxidation of NADH was carried out using ABTS as a mediator, with turnover numbers of 1860 and 93 h$^{-1}$ for the mediator and the cofactor, respectively, rates that were significantly higher than previously reported. A space time yield 1.4 g L$^{-1}$ h$^{-1}$ was achieved for the three-dimensional electrochemical reactor, a value higher than that at a two-dimensional cell [48].

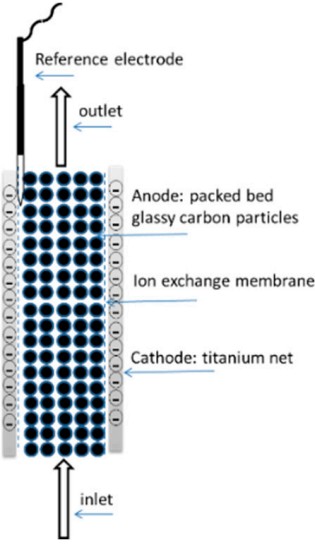

**Figure 6.** Schematic diagram of the three-dimensional electrode [48].

Ruinatscha et al. fabricated a reactor with three-dimensional porous carbon electrodes with a surface area to volume ratio of 19,685 m$^2$m$^{-3}$ (Figure 7).

The reactor was used to produce FADH$_2$, which was then coupled with styrene monooxygenases for the synthesis of styrene oxide. The rate of mass transfer increased in this system and FAD was reduced at a rate of 93 mM h$^{-1}$ producing styrene oxide at a rate of 1.3 mM h$^{-1}$ [144]. Due to the benefits of using microreactors for cofactor regeneration, the direct regeneration of NAD$^+$ was examined. Rodríguez-Hinestroza et al. designed an electrochemical filter press microreactor for the direct anodic regeneration of NAD$^+$, via the horse-liver alcohol dehydrogenase-catalysed synthesis of β-alanine (Figure 8). The platinum and gold electrodes used had 150 microchannels, with a high surface area

of 250 cm$^{-2}$. NADH was directly oxidized at the gold electrode, with a 92% conversion of NADH to NAD$^+$. The produced NAD$^+$ was used for the enzymatic oxidation of carboxybenzyl-β-amino propanol [145].

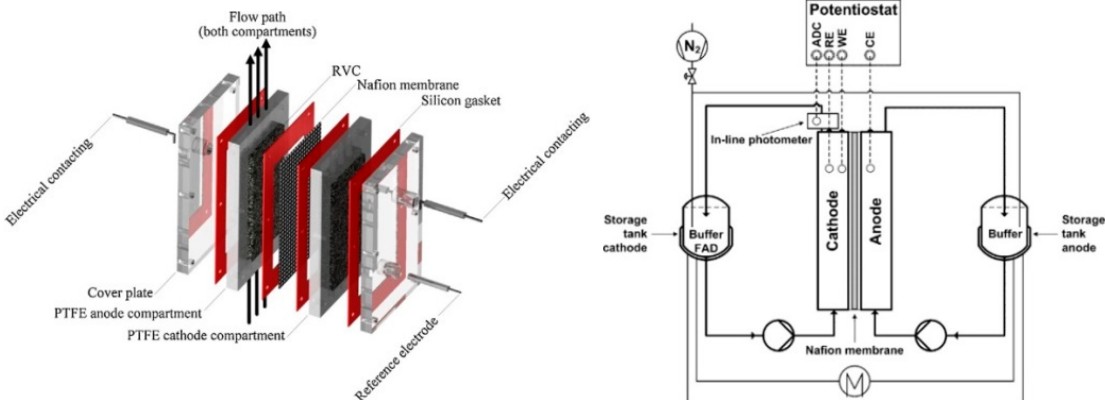

**Figure 7.** Schematic diagram of system for a reactor for the regeneration of FAD. Reprinted from [144] Copyright (2020), with permission from Elsevier.

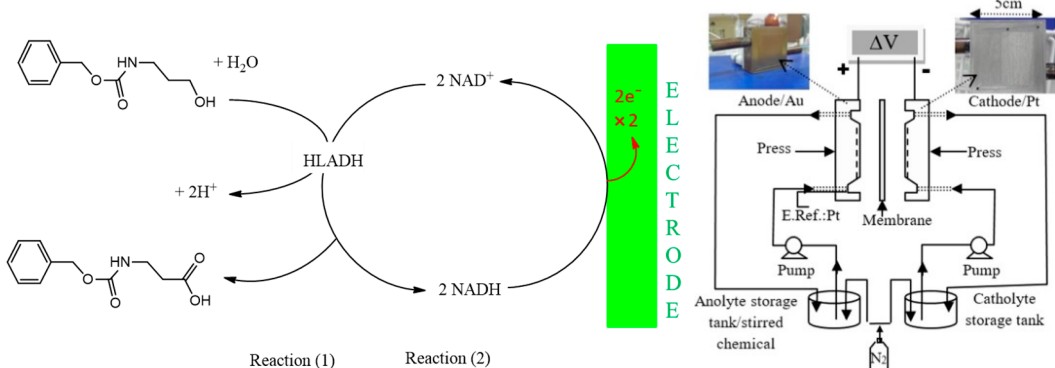

**Figure 8.** Schematic diagram of an electrochemical microreactor system and the electroenzymatic oxidation of carboxybenzyl-β-amino propanol. Reprinted from [145]. Copyright (2020), with permission from Elsevier.

Fisher et al. prepared a multichannel segmented flow bioreactor in which the oxidoreductase pentaerythritol tetranitrate reductase (PETNR) was regenerated electrochemically with the use of methyl viologen as a mediator (Figure 9), replacing NADPH as cofactor. The use of methyl viologen resulted in rates of substrate reduction that were 15–70% of those observed with NADPH [146].

Mazurenko et al. carried out mediated regeneration NAD$^+$ via poly(methylene green) in a flow reactor for the synthesis of D-fructose from D-sorbitol using D-sorbitol dehydrogenase [142]. Electrochemical methods can also be used to immobilise enzymes on the surface of electrodes for subsequent use in a flow reactor. For example; Xiao et al. designed a continuous flow bioreactor based on lipase immobilised by the electrochemical generation of a silica film nanoporous gold-modified glassy carbon electrodes (Figure 10a). In addition, −1.1 V vs. Ag/AgCl was applied on the electrode surface immersed in tetraethoxysilane and lipase solution. This triggered hydroxyl ion production that caused tetraethoxysilane condensation on the electrode surface and lipase was entrapped into silica structure.

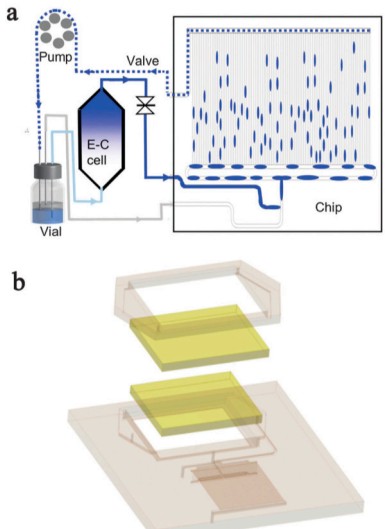

**Figure 9.** (**a**) Schematic diagram of a segmented flow bioreactor system, (**b**) 3D representation of the bioreactor. Reprinted from [146] Copyright (2020), with permission from The Royal Society of Chemistry.

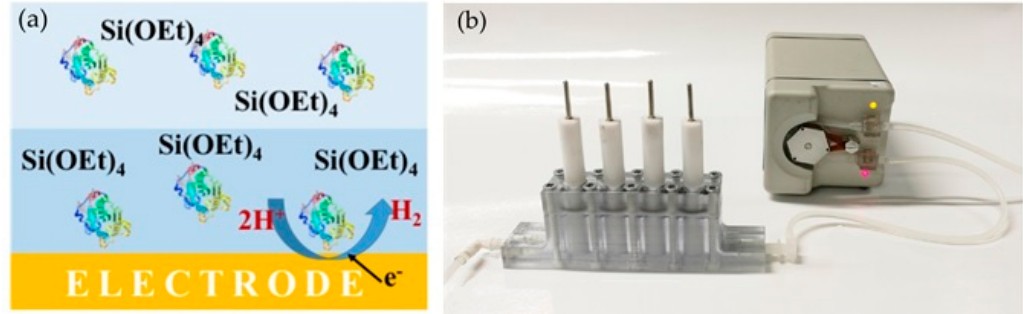

**Figure 10.** (**a**) Schematic diagram of the electrodeposition of a silica layer with entrapped enzyme on an electrode and (**b**) an image of the continuous flow reactor [107].

The modified electrodes were inserted into the flow reactor (Figure 10b) and used to prepare p-nitrophenol from 4-nitrophenyl butyrate (4-NPB), with the full conversion of the substrate (0.075 mM in 2 mL) after eight cycles [107].

## 5. Strategies for Improving Bioelectrocatalysts

### 5.1. High Surface Area Electrodes

One method of improving the rate of reaction in electrochemical bioreactors is to use electrodes with high surface areas to enable high enzyme loadings and consequently higher current densities. Electrodes can be designed with a range of three-dimensional (3D) architectures. Parameters such as the method of enzyme immobilisation, the orientation of the enzyme on the support, the porosity of the support, and diffusional limitations within the pores need to be taken into account [147].

Due to their ease of preparation, porous metal electrodes have been used for a range of applications. For example, dealloyed nanoporous gold (NPG) can be prepared on the electrode surface by electro/chemically dissolving Ag from Au/Ag alloys. The resultant pores have diameters that are sufficiently large to accommodate the enzymes and to enable substrates and products to readily diffuse into and out of the pores. Through changing parameters such as the alloy composition and the dealloying conditions, the pore diameters can vary over the range of 5–700 nm [148]. For example, dealloyed nanoporous gold (NPG) electrodes modified with osmium polymer as a mediator were used

for the determination of glucose and lactose [149]. Dealloyed nanoporous gold (NPG) electrodes have also been used for the direct electron transfer of enzymes such as laccase [69]. The immobilisation of FAD-dependent glucose dehydrogenase, bilirubin oxidases [78] and fructose dehydrogenase [150] were also investigated on porous gold electrodes. Nanoporous gold electrodes are widely used as a biofuel cell [78,151] and biosensor [152]. In the construction of biofuel cells, gold nanoparticles have been attached to the electrode surface to enhance the efficiency of the cells [153,154].

Carbon nanotubes have been widely used to modify electrodes [155–158]. Carbon nanotubes possess properties such as high conductivity and surface areas. A simple method for the modification of electrodes with carbon nanotubes and enzymes is to place a drop of a suspension of enzyme and carbon nanotubes onto the surface of the electrode surface [77]. Electrode modification with carbon nanotubes can facilitate direct electron transfer between the electrode and enzymes such as fructose dehydrogenase [159], or horseradish peroxidase [160], resulting in the preparation of efficient biofuel cells [161,162]. Jourdin et al. used multiwalled carbon nanotubes to improve the performance of a microbial system for the bioreduction of carbon dioxide. Multiwalled carbon nanotube electrodes considerably increased the rate of electron transfer between electrode and microorganisms by 1.65-fold and the acetate production rate by 2.6-fold in comparison with the graphite plate electrodes [163]. The microbial electrosynthesis of acetate from $CO_2$ was described using a vitreous carbon electrode modified with multi-walled carbon nanotubes [164]. Bulutoglu et al. immobilised fused alcohol dehydrogenase on multi-walled carbon nanotubes for the electrocatalytic oxidation of 2,3-butanediol [165]. Bucky papers are flexible, light materials prepared from carbon nanotubes [147,166]. Zhang et al. developed a bioelectrode for electroenzymatic synthesis using a bucky paper electrode on which $[Cp^*Rh(bpy)Cl]^+$ was immobilized as a mediator for the regeneration of NADH. A turnover frequency of $1.3\,s^{-1}$ was achieved for the regeneration of NADH and the system was used for the preparation of D-sorbitol from D-fructose using immobilized D-sorbitol dehydrogenase.

The high porosity of 3D graphene materials can enable higher enzyme loadings, increasing the performance of the electrodes. Choi et al. used a combination of graphitic carbon nitride and reduced graphene oxide as an efficient cathode for the reduction of $O_2$. The $H_2O_2$ produced at the surface of the cathode surface was used for the peroxygenase-catalysed selective hydroxylation of ethylbenzene to (R)-1-phenylethanol [167]. Enzymes can be covalently attached on the graphene hybrid electrodes through the controlled functionalization of graphene causing high stability. Effective enzyme–graphene conjugation can pave the way for direct electron transfer on the electrode surface. Seelajaroen et al. used enzyme–graphene hybrids for the electrochemical preparation of methanol from $CO_2$ using NAD(P)-linked enzymes including formate dehydrogenase, formaldehyde dehydrogenase and alcohol dehydrogenase (Figure 11). Enzymes were covalently bound on to carboxylate-modified graphene surfaces.

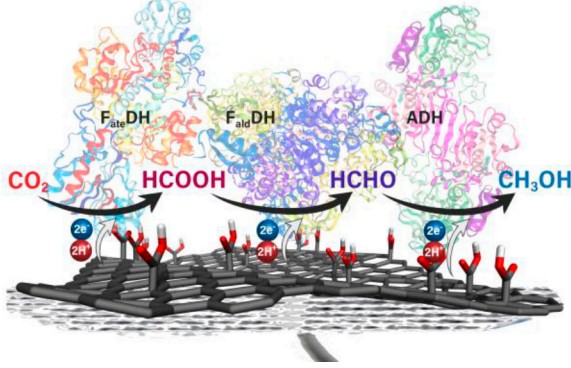

**Figure 11.** The cofactor-free biosynthesis of methanol using formate dehydrogenase, formaldehyde dehydrogenase and alcohol dehydrogenase. Reprinted from [168] Copyright (2020), with permission from the American Chemical Society.

Due to the effective rates of electron transfer from the graphene-modified electrode to the enzymes, considerable improvement was observed with a high faradaic efficiency of 12%. A higher production rate of 0.6 $\mu$mol·h$^{-1}$ was achieved and the current was stable for 20 h. This shows the advantage of using conductive graphene carboxylic acid as support [168].

## 5.2. Enzyme Engineering

Protein engineering can be used to improve the catalytic performance of enzymes [169,170]. Rational design, directed evolution and combinations of these approaches are widely used to prepare more active and stable enzymes [170]. In rational design, mutations are inserted into specific locations in the protein through site-directed mutagenesis. The lack of a full understanding of the relationship between the protein structure and the rate of electron transfer makes it difficult to improve the catalytic activity using rational design. The structural knowledge of proteins is not required for directed evolution and catalytic activity can be improved in the absence of a detailed crystal structure of the enzyme by mimicking Darwinian evolution [170]. Modifying the heme centre of myoglobin increased its dehalogenation activity to over 1000 times that of a native dehaloperoxidase [171]. Recently, Chen et al. prepared cytochrome P450 enzymes through the directed evolution of serine-ligated P450 variants for the preparation of cyclopropene with high efficiency (with a total turnover number (TTN) of up to 5760) and high selectivity (>99.9% ee) [172]. Brandenberg et al. reported a cytochrome P450 variant that was able to catalyse the C2-amidation of indole. Both the heme and reductase domains were modified, improving the catalytic activity with the total turnover number increasing from 100 to 8400 and the product yield from 2.1 to 90% [173]. Mateljak et al. used computational design with directed evolution to design fungal high-redox-potential laccases that exhibited high stability and activity toward redox mediators [174]. Further work was performed to investigate the use of these laccases for the reduction of O$_2$ in the presence and absence of ABTS. The designed laccases could reduce O$_2$ at low overpotentials [175]. Protein engineering has been used to adjust the properties of NAD(P)H-dependent oxidoreductases [176]. For example, Liu et al. used directed evolution to prepare an NADH-dependent alcohol dehydrogenase from *Lactococcus lactis* for the production of isobutanol. The catalytic efficiency of the engineered enzyme increased by a factor of 160 in comparison with the wild-type enzyme [177]. Li et al. used directed evolution to improve the catalytic activity of puritative oxidreductase in the production of 1,3-propanediol [178].

## 5.3. Enzyme Cascades

Typically, a biocatalytic cascade reaction is a reaction system in which two or more transformations are performed simultaneously [179]. Through biocatalytic cascades, the isolation of reaction intermediates is circumvented, saving time and the use of reagents. The synthesis of chemicals with unstable intermediates is feasible as the isolation of the intermediates is not necessary [180]. Enzyme cascades can enhance the current density as the oxidation of the substrate in a sequential manner enables more electrons to be extracted [34]. Dong et al. designed a biphasic bioelectrocatalytic system that used a reaction cascade to prepare (R)-ethyl-4-cyano-3-hydroxybutyrate from ethyl 4-chloroacetoacetate using alcohol dehydrogenase and halohydrin dehalogenase (Figure 12). Ethyl 4-chloroacetoacetate was reduced to (S)-4-chloro-3-hydroxybutanoate using alcohol dehydrogenase which was then converted to its R enantiomer using halohydrin dehalogenase. Efficient NADH regeneration was performed using diaphorase with the redox polymer cobaltocene-modified poly-(allylamine) as a mediator on the electrode surface [181]. In total, 85% of substrate was converted into (R)-ethyl-4-cyano-3-hydroxybutyrate, an 8.8 higher yield than that achieved with a single-phase system.

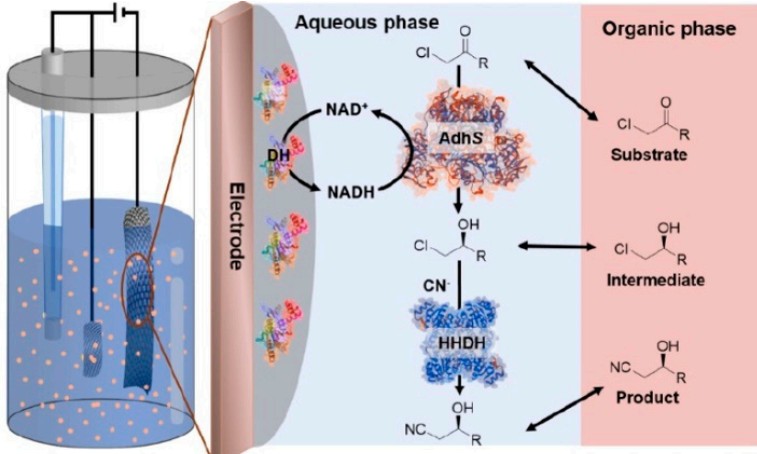

**Figure 12.** Schematic diagram of the biphasic bioelectrocatalytic reaction cascade for the synthesis of (R)-ethyl-4-cyano-3-hydroxybutyrate. Reprinted from [181] Copyright (2020) with permission from American Chemical Society.

Abdellaoui et al. combined aldehyde deformylating oxygenase, immobilised on the surface of an electrode, with NAD$^+$-dependent alcohol dehydrogenase to create an enzymatic cascade reaction for the conversion of fatty alcohols to aldehydes followed by the decarbonylation to produce alkenes (Figure 13) [182].

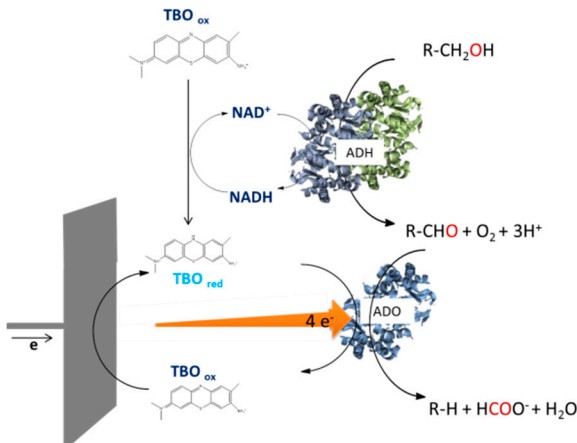

**Figure 13.** Schematic diagram of a bioelectrochemical enzymatic cascade for the preparation of alkenes [182].

Kuk et al. constructed a photoelectrochemical system for the production of methanol by a multienzyme cascade (a three-dehydrogenase cascade system) and the efficient regeneration of the NADH via visible-light assistance (Figure 14) [183].

Chen et al. developed an enzyme cascade to produce chiral amines from nitrogen. NH$_3$ was produced from N$_2$ via nitrogenase and then used, with L-alanine dehydrogenase, to produce alanine from pyruvate. The desired chiral amines were produced via the ω-transaminase transfer of an amino group from alanine. Pyruvate was generated as a by-product and converted to alanine using L-alanine dehydrogenase. The system was used for the successful amination of the substrates, 4-phenyl-2 butanone, 4-methyl methoxy phenyl acetone, phenoxyacetone, 2-pentanone, methoxyacetone and 2-octanone with an enantiomeric excess >99% (Figure 15) [52].

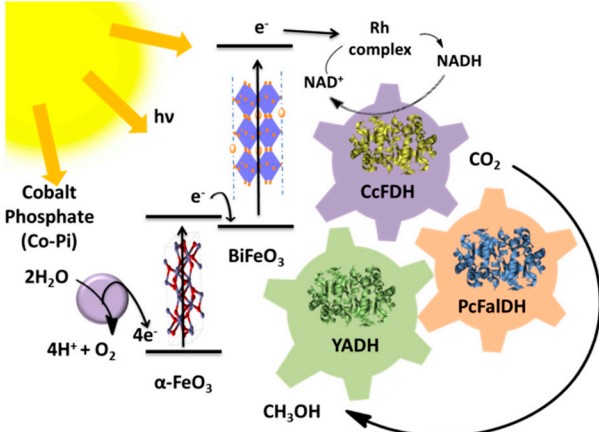

**Figure 14.** Schematic diagram of the light-assisted synthesis of methanol from $CO_2$ via an enzyme cascade (formaldehyde dehydrogenase, formate dehydrogenase, and alcohol dehydrogenase) [183].

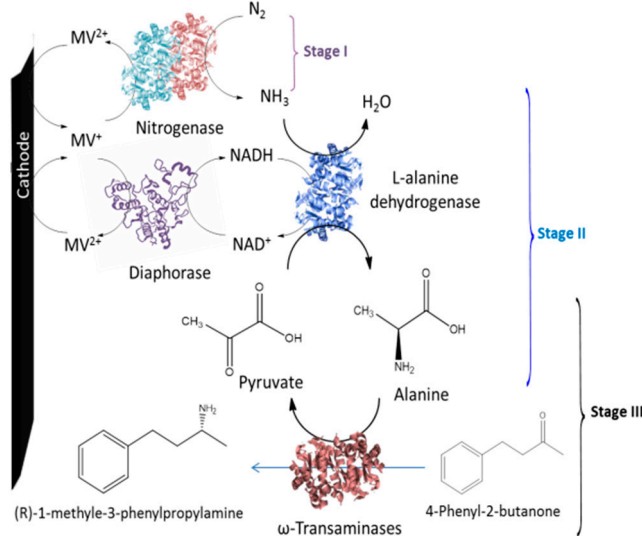

**Figure 15.** A schematic of bioelectrochemical asymmetric amination [52].

In a sequential catalytic cascade, noble metal catalysts can also be used. For example, the biocatalytic reduction of nitrite to ammonia is slow in comparison with metal catalysts [184]. Duca et al. fabricated a hybrid electrochemical cascade that combined nitrate reductase and Pt nanoparticles on a carbon electrode surface. The reduction of nitrate to nitrite and nitrite to ammonia was carried out by the Pt nanoparticles and nitrate reductase, respectively [185].

## 6. Bioelectrosynthesis

A variety of oxidoreductases have been used for the asymmetric synthesis of amines, amino acids and alcohols [186–188]. Dong et al. developed a biphasic bioelectrocatalysis system for the synthesis of (R)-CHCN (81.2%), (R)-3-hydroxy-3-Phenylpropanenitrile (96.8% ee), (S)-3-hydroxy-4-phenylbutanenitrile (94.6% ee) using alcohol dehydrogenase and halohydrin dehalogenase. The system showed high selectivity for the synthesis of chiral β-hydroxy nitriles. The NADH cofactor was efficiently regenerated using diaphorase and a cobaltocene-modified poly(allylamine) redox polymer [181]. Combining enzymatic electrosynthesis with biofuel cells can pave the way for the self-powered bioelectrosynthesis of valuable chemicals. Wu et al. designed an electroenzymatic system by combining a bioelectrosynthesis cell and an enzymatic fuel cell to prepare L-3,4-dDihydroxyphenylalanine with a coulombic efficiency of 90% [189]. Chen et al. combined a $H_2/\alpha$-keto acid enzymatic fuel cells with bioelectrosynthesis to synthesis chiral amino acids with high efficiency. An enzymatic cascade

consisting of nitrogenase, diaphorase and L-leucine dehydrogenase was used at the cathode to convert nitrogen to chiral amino acids [190]. The $NH_3$ and NADH prepared in the reaction were used for the synthesis of L-norleucine from 2-ketohexanoic acid via leucine dehydrogenase (LeuDH). A high $NH_3$ conversion ratio (92%) and a high faradaic efficiency (87.1%) was observed.

Carbon dioxide emissions are increasing, dramatically bringing about an environmental crisis. Therefore, electrochemical $CO_2$ reduction to useful chemicals has gained great interest to reduce $CO_2$ emissions. Cai et al. reduced $CO_2$ to ethylene and propene via the VFe protein of vanadium nitrogenase that was contacted with the electrode mediated by cobaltocene/cobaltocenium (Figure 16). A study represented a new approach for preparing C–C bonds through a single metalloenzyme [191].

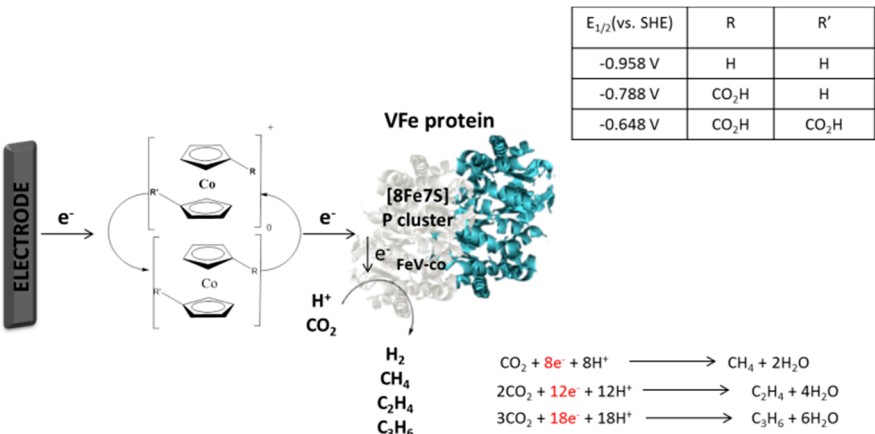

**Figure 16.** A schematic of C–C bond preparation via bioelectrocatalysts, Amperometric i−t trace carried out for enzymatic electroreduction of $CO_2$ via VFe nitrogenase [191].

Yuan et al. reported $CO_2$ reduction producing formate with a high faradaic efficiency of 99% at a low potential −0.66 V vs. The standard hydrogen electrode (SHE). They used immobilised molybdenum-dependent formate mediated by cobaltocene on the electrode surface [192]. In some studies, $CO_2$ reduction was carried out via enzymatic cascade, using formate dehydrogenases, formaldehyde dehydrogenase and alcohol dehydrogenase to produce $CH_3OH$ [193]. Bioplastics can be prepared from $CO_2$ reduction via bioelectrocatalysts. For example, Sciarria et al. synthesized polyhydroxyalkanoates (PHA) from $CO_2$ through a bioelectrochemical reactor. $CO_2$ was converted into bioplastics successfully in a way that 0.41 kg of carbon as PHA was achieved per 1 kg of $C_{CO2}$ in the whole system [194].

Nitrogenase catalyses the reduction of $N_2$ to ammonia. Nitrogenase can also be used for the reduction of CO, nitrite, azide and cyanide that can then be used for the synthesis of a range of products [195]. Lee et al. wired nitrogenase to a redox-polymer (neutral red-modified poly(glycidyl methacrylate-co-methylmethacrylate-co-poly(ethyleneglycol)methacrylate)) for the conversion of nitrogen to ammonia. [196]. Milton et al. reported on the use of nitrogenase to prepare ammonia from azide and nitrite using cobaltocene as a mediator. The system operated at a potential of −1.25 V with 70 and 234 nmol of $NH_3$ produced from the reduction of $N_3^-$ and $NO_2^-$, respectively [197]. An ATP-free-mediated electron-transfer system operating at a potential of −0.58 V was developed by Lee et al. for the synthesis of ammonia using cobaltocene-functionalized poly(allylamine) (Cc-PAA) as a mediator. The amount of ammonia prepared was 30 ± 5 and 7 ± 2 nmol from the reduction of $N_3^-$ and $NO_2^-$, respectively [198].

## 7. Conclusions

Oxidoreductases catalyse redox reactions and have been successfully used to prepare biofuel cells and biosensors. The enzymes undergo electron transfer with the surface of the electrode via mediated or direct electron transfer mechanisms. However, relatively few oxidoreductases can undergo

direct electron transfer because their redox active centres are placed deep within their structures. While enzymes have been widely used in batch and flow reactors, the use of oxidoreductases in bioreactors has been mainly confined to cofactor regeneration. A number of strategies that entail the use of high surface area electrodes, enzyme engineering and enzyme cascades have been used to improve the performance of bioelectrocatalysts. As an example of the scope of bioelectrocatalysts, a variety of oxidoreductases has been used in asymmetric synthesis reactions. While the scope of bioelectrocatalysts can be broadened, the successful use of bioelectrocatalysts will lie initially in the development of small-scale reaction systems that focus on high-value fine chemicals. The scale up of the reaction systems to the size required for bulk manufacture will be a significant challenge. Due to this challenge, the development of electrocatalytic bioreactors will likely focus on the preparation of products at a small scale taking advantage of the relatively low-cost systems that can be assembled in a relatively simple and rapid manner. Such systems are adaptable and are particularly suited for systems that may be required in process optimization or in the preparation of products or intermediates on a small (g) scale. Screening the performance of mutant enzymes is also more feasible given the relatively small amounts of enzymes required in such reactors. A particular area of interest lies in the use of multiple enzymes for cascade reactions, where rapid prototyping is particularly useful in screening and selecting the optimal operational conditions.

**Author Contributions:** Writing, S.A.; editing and review, M.N.-A.; writing—review and editing, E.M. All authors have read and agreed to the published version of the manuscript.

**Funding:** This research was funded by the Science Foundation Ireland (SFI) Research Centre for Pharmaceuticals under Grant Number 12/RC/2275.

**Conflicts of Interest:** The authors declare no conflict of interest.

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
