# Peer review of "Enzymatic Bioreactors: An Electrochemical Perspective"

_catalysts, doi:10.3390/catal10111232_

Round 1
Reviewer 1 Report
This is one of the many review papers that concern the subject of biomaterials that are used to construct biocatalytic reactors. The authors describe mechanisms of electron transfer, methods of immobilising enzymes, properties and application of biocatalytic reactors, batch and flow reactors, application of polymers, nanomaterials, enzymes cascades in the construction of electrodes. This publication is appropriate for Catalysts. The manuscript is well written and structured. The figures are clear, demonstrating the concepts proposed by the authors. I therefore fully support publication of this original work after minor corrections.
1. Very often in the construction of bioelectrodes (both bioanodes and biocathodes) gold nanoparticles are used, among others to improve transport efficiency. Various groups use them in their work, e.g. Shleev and Bilewicz groups. I am asking for a short development of this topic in the publication. I recommend to add literature references
Mediatorless sugar/oxygen enzymatic fuel cells based on gold nanoparticle - modified electrodes, X. Wang, M. Falk, R. Ortiz, H. Matsumura, J. Bobacka, R. Ludwig, M. Bergelin, L. Gorton, S. Shleev, Biosensors and Bioelectronics 31 (2012) 219 – 22, doi:10.1016/j.bios.2011.10.020
Multi-substrate biofuel cell utilizing glucose, fructose and sucrose as the anode fuels, M. Kizling, M. Dzwonek, A. Nowak, Ł. Tymecki, K. Stolarczyk, A. Wieckowska, R. Bilewicz, Nanomaterials 10(8) (2020) 1534, doi: 10.3390/nano10081534
and others
2. Please note the text editing:
- remove double spaces
- In line 224 50@m please change to 50nm
- In line
344 Figure 10. (a) Schematic diagram of the electrodeposition of a silica layer with entrapped enzyme on
345 an electrode and (b) image of the continuous flow reactor [105]
please change to
344 Figure 10. (A) Schematic diagram of the electrodeposition of a silica layer with entrapped enzyme on
345 an electrode and (B) image of the continuous flow reactor [105]
Author Response
Reviewer 1
- Very often in the construction of bioelectrodes (both bioanodes and biocathodes) gold nanoparticles are used, among others to improve transport efficiency. Various groups use them in their work, e.g. Shleev and Bilewicz groups. I am asking for a short development of this topic in the publication. I recommend to add literature references
Mediatorless sugar/oxygen enzymatic fuel cells based on gold nanoparticle - modified electrodes, X. Wang, M. Falk, R. Ortiz, H. Matsumura, J. Bobacka, R. Ludwig, M. Bergelin, L. Gorton, S. Shleev, Biosensors and Bioelectronics 31 (2012) 219 – 22, doi:10.1016/j.bios.2011.10.020
Multi-substrate biofuel cell utilizing glucose, fructose and sucrose as the anode fuels, M. Kizling, M. Dzwonek, A. Nowak, Ł. Tymecki, K. Stolarczyk, A. Wieckowska, R. Bilewicz, Nanomaterials 10(8) (2020) 1534, doi: 10.3390/nano10081534
and others
The focus of the manuscript is on biocatalysis rather than on biofuel cells. However as suggested by the reviewer, gold nanoparticles can be utilised to obtain higher efficiencies. The references mentioned by the reviewer are now included in the reference section and the following text has been included “In the construction of bioelectrodes, gold nanoparticles have been attached to the electrode surface to improve the efficiency of the cells [153,154]” (line 366-367, page 12)
- Please note the text editing:
- remove double spaces:
Double spaces have been removed.
- In line 224 50@m please change to 50 nm:
(Line 222, page 7) It has been corrected “50 nm”
- In line 344 Figure 10. (a) Schematic diagram of the electrodeposition of a silica layer with entrapped enzyme on 345 an electrode and (b) image of the continuous flow reactor [105] please change to: 344 Figure 10. (A) Schematic diagram of the electrodeposition of a silica layer with entrapped enzyme on 345 an electrode and (B) image of the continuous flow reactor [105].
In the catalysts template file, letters in sub-figures are not capitalised
Reviewer 2 Report
This manuscript is a comprehensive and extensive review with focus on bioelectrocatalysis-based electrochemical synthesis. This is acceptable for publication in Catalysis. However, the authors may consider the following point for improving the manuscript and for the authors’ further research.
1) The authors may discuss overpotential issue even for bioelectrocatalytic synthesis. In this sense, the direct and ABTS-catalyzed oxidation of NADH might not be good example for bioelectrosynthesis.
2) In section 4.2.2, there is no example of oxidoreductase for monolithic reactors. Immobilization issue of other enzymes is welcome, but the authors should give any short comment on this matter.
3) The authors should distinguish between NAD(P)-dependent enzymes and NAD(P)-linked enzymes (or sometimed called NAD(P)-reducing enzymes). The former strictly utilizes NAD(P)(H), but the latter utilizes several redox compound in place of NAD(P)(H). The enzymes used in Fig, 11 (Ref. 164) might be assigned to NAD(P)-linked enzymes. Some comments may be useful for readers.
Minor points
p.6, L. 178; “increases can increase” should be “can increase”.
p.7, L.244; 50 “micro”m?
Author Response
Reviewer 2:
- The authors may discuss overpotential issue even for bioelectrocatalytic synthesis. In this sense, the direct and ABTS-catalyzed oxidation of NADH might not be good example for bioelectrosynthesis.
(Line 65-66, page 2) As mentioned in the manuscript, the direct regeneration of nicotinamide coenzymes at unmodified electrode surfaces requires the use of high overpotentials. (Line 301-303, page 9) Moreover, a study related to ABTS-catalyzed oxidation of NADH has been mentioned in electrochemical reactors to cover all studies carried out in the electrochemical bioreactors.
- In section 4.2.2, there is no example of oxidoreductase for monolithic reactors. Immobilization issue of other enzymes is welcome, but the authors should give any short comment on this matter.
(Line 250-252, Page 7-8) an example of oxidoreductases used in monolithic reactors has been included in the manuscript. The text has been added; “Logan et al. immobilized three enzymes (invertase, glucose oxidase, and horseradish peroxidase) on polymer monoliths in different regions of a microfluidic device”.
- The authors should distinguish between NAD(P)-dependent enzymes and NAD(P)-linked enzymes (or sometimes called NAD(P)-reducing enzymes). The former strictly utilizes NAD(P)(H), but the latter utilizes several redox compound in place of NAD(P)(H). The enzymes used in Fig, 11 (Ref. 164) might be assigned to NAD(P)-linked enzymes. Some comments may be useful for readers.
(line 394, Page 13) The enzymes used in Fig, 11 have been described as NAD(P)-linked enzymes as recommended. The text has been added; “NAD(P)-linked enzymes including formate dehydrogenase, formaldehyde dehydrogenase and alcohol dehydrogenase”.
- Minor points
- 6, L. 178; “increases can increase” should be “can increase”.
(Line 176, page 6) The text has been corrected; “The use of flow can increase rates of mass transfer”
- 7, L.244; 50 “micro”m?
(Line 222, page 7) The text has been corrected; “50 nm”
Reviewer 3 Report
This paper presents an interesting and topical review on enzymatic reactors with an electrochemical respective. There are several minor corrections to consider and an important major correction / re-write of the conclusion.
The following comments should be considered prior to eventual publication:
- the title should be more specific (relevant) e.g. enzymatic reactors rather than simply bioreactors. Bioreactors includes other types such as microbial bioreactors, but these were not reviewed.
- Section 2 intro. A good reference by Minteer on DET (and MET) that might be included that covers several enzyme types https://doi.org/10.1098/rsif.2017.0253. The ET distance is critical and also valuable information that may be added.
- Section 2.1. Other mediator characterstics are important and should be detailed e.g. reversibility and affinity. The potential may be good but not all mediators with good potentials work.
- P2 l67, the use of mediators in the text is a bit confusing. The mediators are not used as mediators but as electrocatalysts?
- Section 2.1. It would be beneficial to clarify/explain a bit more the link between NAD+, NAD, NADP+, NADP
- P3 l68, bracket missing before (ABTS).
- For turnover units, somtimes "per hours" sometimes "h-1", this needs to be homogenized
- Throughout the document there are formatting issues with spaces (sometimes missing or too many between words and before brackets) and sub/superscripts
- Inconsistencies UK and US spelling e.g. immobilization vs immobilise
- P5 l147. A nice citation but the explanation is not clear. It could be re-written, something like " COOH groups were attached via diazonium surface chemistry then used for...coupling the enzyme "
- l54. Section AA? What is AA?
- 160. The text should concern redox polymers but reference 85 concerns sol-gel matrices not redox polymers.
- Fig 3. Is it possible to add more scientific details such as dimensions or materials?
- l224. there is a unit issue
- l237. I did not understand the relation between enzyme immobilisation and the need for high backpressure?
- l264. What material are the nanosprings made of?
- FAD abbreviation, requires full name to be specified
- l310. The authors report a value but it is not a surface area. Its a surface area to volume ratio (not the same)
- Figure 10. Remove arrows or provide text labels
- l373. Ref 156 refers to galactate oxidase not glucose oxidase? I would double check that the DET with GOx is reliable in ref 157. I didn't check if the enzyme is deglycoslyated.
- Section 5.1. For the section concerning carbon nanotubes, I suggest to mention carbon nanotube buckypaper. An high surface area material for DET and MET that has been used, at least once, for biosythesis of D-sorbitol, and could be interesting for future applications. Example relevant references: https://doi.org/10.1002/cctc.201800681; https://doi.org/10.1039/C8EE00330K
- Section 5.2. Are there any examples of engineered NAD dependant enzymes? NAD dependant enzymes most relevant
- l505, l507. Azide or nitrate? It should be NO3-?
- Conclusion - after the nice review the conclusion should give a new insight or critical view. For example, the authors suggest small scale experimants using relatively low cost systems - such as (are certain approach or approaches better suited for low cost)? A particular interest lies in cascade reactions - why and perhaps only in certain cases when one enzyme is not sufficient? a variety of oxidoreductases have been used in asymmetric synthesis - what lessons have learned - is this the most promising future direction, etc?
Author Response
Reviewer 3
- the title should be more specific (relevant) e.g. enzymatic reactors rather than simply bioreactors. Bioreactors includes other types such as microbial bioreactors, but these were not reviewed.
(line 2, page 1) The title has been made more specific as suggested by the reviewer to: “Enzymatic bioreactors; an electrochemical perspective”
- Section 2 intro. A good reference by Minteer on DET (and MET) that might be included that covers several enzyme types https://doi.org/10.1098/rsif.2017.0253. The ET distance is critical and also valuable information that may be added.
(line 104-106, Page 4) The importance of distance electron transfer was discussed, e.g line 41, page 2. The recommended reference has been added as reference [24].
- Section 2.1. Other mediator characterstics are important and should be detailed e.g. reversibility and affinity. The potential may be good but not all mediators with good potentials work.
(line 61, Page 2) As described in reference [33] properties such as stability, selectivity and reversibility are also required characteristics of mediator species. (line 60, Page 2) The following text has been included in the manuscript “Properties such as mediator stability, selectivity and the electrochemical reversibility of the redox couple also need to be considered”
- P2 l67, the use of mediators in the text is a bit confusing. The mediators are not used as mediators but as electrocatalysts?
(line 49-50, Page 2) The mediators act as electron shuttles, not as electrocatalysts.
- Section 2.1. It would be beneficial to clarify/explain a bit more the link between NAD+, NAD, NADP+, NADP
(Line 65 and 67, page 2) This has been clarified as suggested by the reviewer, NAD(P)+ refers to the oxidised form of the cofactors, all uses of NAD(P) have been changed to NAD(P)+.
- P3 l68, bracket missing before (ABTS).
(Line 69, page 3) The bracket has been included; “(ABTS)”
- For turnover units, sometimes "per hours" sometimes "h-1", this needs to be homogenized
(Line 71, page 3-line 302, page 9) per hour has been changed to h-1 as recommended.
- Throughout the document there are formatting issues with spaces (sometimes missing or too many between words and before brackets) and sub/superscripts
The document has been checked for formatting issues.
- Inconsistencies UK and US spelling e.g. immobilization vs immobilise
The use of such spelling has been checked in the document. The format has been changed to the UK format.
- P5 l147. A nice citation but the explanation is not clear. It could be re-written, something like " COOH groups were attached via diazonium surface chemistry then used for...coupling the enzyme "
(Line 143-146, page 5) The text has been modified “The immobilisation of bilirubin oxidases on nanoporous gold electrodes is carried out. COOH groups were attached via diazonium surface chemistry on the electrode surface then they were used for covalently coupling bilirubin oxidases”.
- Section AA? What is AA?
(Line 151, page 5) the text has been replaced with (section 2.1) in which redox polymers are discussed.
- The text should concern redox polymers but reference 85 concerns sol-gel matrices not redox polymers.
(Line 157-160, page 5) this reference has been replaced with a new reference [86]. The following text has been added “As an example of a biocatalytic system, Alsaoub et al. constructed a biosupercapacitor using an Os complex modified redox polymer to immobilize glucose oxidase and flavin adenine dinucleotide (FAD)‐dependent glucose dehydrogenase at the anode and bilirubin oxidase at the cathode [86].
- Fig 3. Is it possible to add more scientific details such as dimensions or materials?
(Figure 3, page 7) It is only a simple schematic diagram of reactors; more details related to each type of reactors are available in sections that they are described: “4.2.1. Packed-bed reactors (line 218-237), 4.2.2. Monolithic reactors (line 238-257), 4.2.1. Wall-coated reactors (line 258-278)”
- there is a unit issue
The issue has been checked.
- I did not understand the relation between enzyme immobilisation and the need for high backpressure?
(Line 235, page 7) The sentence has been clarified “However, using beads in the reactors has disadvantages”
- What material are the nanosprings made of?
(Line 264, page 8) As described in reference [131] the nanosprings are made of silicon dioxide.
- FAD abbreviation, requires full name to be specified
(Line 120-121, page 4) The full name of FAD has been added “flavin adenine dinucleotide”
- The authors report a value but it is not a surface area. Its a surface area to volume ratio (not the same)
(Line 308-309, page 10) Surface area has been replaced with surface area to volume ratio as recommended.
- Figure 10. Remove arrows or provide text labels
(Figure 10, page 11) As suggested by the reviewer, the arrows were removed.
- Ref 156 refers to galactate oxidase not glucose oxidase? I would double check that the DET with GOx is reliable in ref 157. I didn't check if the enzyme is deglycoslyated.
(Line 365, page 12) The reviewer is correct and references 156 and 157 have been removed.
- Section 5.1. For the section concerning carbon nanotubes, I suggest to mention carbon nanotube buckypaper. An high surface area material for DET and MET that has been used, at least once, for biosythesis of D-sorbitol, and could be interesting for future applications. Example relevant references: https://doi.org/10.1002/cctc.201800681; https://doi.org/10.1039/C8EE00330K add this ref but state for biofuel cells
(Line 381-385, Page 12) As recommended, buckypapers have been included in the manuscript. The following text has been included; “Buckypapers are flexible, light materials prepared from carbon nanotubes [147,166]. Zhang et al. developed a bioelectrode for electroenzymatic synthesis using a bucky paper electrode on which [Cp*Rh(bpy)Cl]+ was immobilized as a mediator for the regeneration of NADH. A turnover frequency of 1.3 s−1 was achieved for the regeneration of NADH and the system used for the preparation of D‐sorbitol from D‐fructose using immobilized D‐sorbitol dehydrogenase”
- Section 5.2. Are there any examples of engineered NAD dependant enzymes? NAD dependant enzymes most relevant
(Line 424-429, Page 14) engineered NAD(P)H dependant enzymes were included in the manuscript. The following text has been added; “ Protein engineering has been used to adjust the properties of NAD(P)H-dependent oxidoreductases [176]. For example, Liu et al. used directed evolution to prepare an NADH-dependent alcohol dehydrogenase from Lactococcus lactis for the production of isobutanol. The catalytic efficiency of the engineered enzyme increased by a factor of 160 in comparison with the wild-type enzyme [177]. Li et al. used directed evolution to improve the catalytic activity of puritative oxidreductase in the production of 1,3-propanediol [178]”
- l505, l507. Azide or nitrate? It should be NO3-?
(Line 515-518, page 17) In mentioned study, MoFe protein bioelectrode was used for the reduction of N3− to NH3, and NO2− to NH3 using an electron mediator.
- Conclusion - after the nice review the conclusion should give a new insight or critical view. For example, the authors suggest small scale experimants using relatively low cost systems - such as (are certain approach or approaches better suited for low cost)? A particular interest lies in cascade reactions - why and perhaps only in certain cases when one enzyme is not sufficient? a variety of oxidoreductases have been used in asymmetric synthesis - what lessons have learned - is this the most promising future direction, etc?
(line 534-539, page 18) The following text has been added; “. Such systems are adaptable and are particularly suited for one of systems that may be required in process optimization or in the preparation of products or intermediates on a small (g) scale. Screening of the performance of mutant enzymes is also more feasible given the relatively small amounts of enzymes required in such reactors. A particular area of interest lies in the use of multiple enzymes for cascade reactions, where rapid prototyping is particularly useful in screening and selecting the optimal operational conditions”